# Clonal Hematopoiesis and Outcomes After High-Dose Chemotherapy and Autologous Stem Cell Transplantation in Patients with AML, Myeloma, and Lymphoma

**DOI:** 10.3390/ijms26168021

**Published:** 2025-08-19

**Authors:** Corinne Natalie Schmid, Katharina Sponagel, Ulrike Bacher, Katja Seipel, Naomi Porret, Gertrud Wiedemann, Michèle Hoffmann, Michael Daskalakis, Thomas Pabst

**Affiliations:** 1Department of Medical Oncology, Inselspital, University Hospital of Bern, 3010 Bern, Switzerland; corinne.schmid@students.unibe.ch (C.N.S.); katharina.sponagel@students.unibe.ch (K.S.);; 2Department of Hematology and Central Hematology Laboratory, Inselspital, Bern University Hospital, University of Bern, 3010 Bern, Switzerlandnaomi.porret@insel.ch (N.P.);; 3Department for BioMedical Research, University of Bern, 3010 Bern, Switzerland

**Keywords:** clonal hematopoiesis (CH), acute myeloid leukemia (AML), lymphoma, myeloma, autologous stem cell transplantation (ASCT), high-dose chemotherapy (HDCT)

## Abstract

Autologous stem cell transplantation (ASCT) after high-dose chemo-therapy (HDCT) is an option of consolidation therapy in patients with AML, lymphoma, or myeloma. Clonal hematopoiesis (CH) is a premalignant state, associated with an increased risk of hematological cancer. The incidence of CH in patients with AML, myeloma, and lymphoma and its effect on the outcome after HDCT/ASCT remain poorly studied. Here we screened 142 patients treated with HDCT/ASCT between 2002 and 2021 at Bern University Hospital for somatic gene mutations in *ASXL1*, *DNMT3A*, *JAK2*, *TET2*, and *TP53*. CH-associated somatic gene mutations were detected in 14/31 AML patients (45%), 13/64 myeloma patients (20%), and 9/47 lymphoma patients (19%). Clinical characteristics, treatment modalities, and responses to treatment were similar in patients with and without CH. Patients with CH-associated gene mutations had higher relapse rates and reduced progression free survival, most evident in lymphoma patients (*p* = 0.007). Overall survival tended to be shorter in lymphoma patients with CH-associated mutations (*p* = 0.078), whereas this was not observed in AML and myeloma patients. Survival in lymphoma patients with CH was inferior, which may have an impact on post-transplant surveillance strategies in the future. In contrast, survival outcomes were not associated significantly with CH in AML and myeloma patients in our study. Longer follow-ups and larger cohorts will be needed to validate our observations.

## 1. Introduction

HDCT followed by ASCT is performed in patients with aggressive lymphoma types and in patients with myeloma for consolidation in early lines, and for consolidation in patients with favorable- and intermediate-risk AML in first complete remission [1]. In the past decade, clonal hematopoiesis of indeterminate potential (CH) became a focus of interest in this field following the introduction of next-generation sequencing (NGS) technology. According to numerous studies investigating large-sized cohorts, the incidence of CH increases with age, being detectable in approximately 1% of humans < 50 years of age, but increasing to 10% in individuals over the age of 65 years [2]. Strikingly, in individuals with some cancer subtypes, the incidence of CH increases to up to 33% in patients aged over 70 years [3]. CH is defined as the presence of a somatic mutation associated with hematological malignancies, such as *ASXL1*, *DNMT3A*, *JAK2*, *TET2*, and *TP53*, without morphologic or clinical evidence of a hematological neoplasm [4,5]. The risk of developing a hematological neoplasia in individuals with CH is 0.5–1% per year with a twelve-fold increased risk compared to individuals without CH [6]. Also, the risk of developing a secondary myeloid neoplasm after treatment of solid tumors or lymphomas is increased [3,7,8,9]. Beyond that, CH increases the risk for inflammation and conditions such as cardiovascular, chronic liver, and kidney diseases, as well as osteoporosis and gout [2,4,10,11,12]. In contrast, individuals with CH were reported to have a lower risk of developing Alzheimer’s disease [13].

However, open questions remain regarding the impact of clonal hematopoiesis on the clinical course in individuals with various hemato-oncologic malignancies. In this study we focused on patients with AML, myeloma, and lymphoma aiming to evaluate the frequency of clonal hematopoiesis in these patients and to investigate a possible impact on outcomes and response to HDCT/ASCT.

## 2. Results

### 2.1. Patient and Disease Characteristics

A total of 142 patients with hematological malignancies undergoing HDCT/ASCT were included in this study, comprising 64 myeloma, 47 lymphoma, and 31 AML. One hundred days after ASCT, gene mutations associated with CH were detected by NGS in peripheral blood monocytes (PBMC) of 36 patients (25%), including 14 AML (14/31, 45%), 9 lymphoma (9/47, 19%), and 13 myeloma (13/64, 20%). A total of 49 gene mutations were detected in 142 patients with a median VAF of 16% (range 1–54%) including 22 *DNMT3A*, 19 *TET2*, and 3 *TP53* gene mutations (Appendix A).

Clinical characteristics were evaluated by univariate analysis comparing subgroups of patients with and without CH (Table 1). Median age (63.5 years in the CH group, 60.5 years in the noCH group) and sex distribution did not differ in the CH and noCH groups. In myeloma patients, there was no difference in staging, high-risk cytogenetics, and the occurrence of complications, particularly regarding renal function and hypercalcemia, in the two myeloma subgroups. In lymphoma patients, there was no difference in staging in the two groups with the majority of patients with stage IV disease. There was one notable difference in angioimmunoblastic T-cell lymphoma (AITL), where all patients were positive for CH (*p* = 0.005). The AML FAB classification was distributed equally between the CH and noCH groups. Disease-related gene mutations in *FLT3* and *NPM1* were equally distributed in both AML subgroups. Disease-related gene mutations in *IDH2* were detected in PBMC of six AML patients, *SRSF2* mutations in five AML patients, and *NRAS* mutations in three AML patients.

### 2.2. Clonal Hematopoiesis

*ASXL1*, *DNMT3A*, *TET2*, and *TP53* gene mutations associated with clonal hematopoiesis (CH) were detected in peripheral blood monocytes of 36 patients (36/142, 25%) with variant allele frequencies (VAFs) of median 18%, range 1–54% (Appendix A). *JAK2* mutations were not detected; *TP53* gene mutations were restricted to three lymphoma patients with a median VAF 11%; range 10–52% (*p* = 0.0074); *ASXL1* gene mutations were restricted to two AML patients (Appendix A). *DNMT3A* or *TET2* gene mutations were detected in AML, lymphoma, or myeloma patients, with equal prevalence. In AML patients, *DNMT3A* mutations were present with a VAF range of 2–47% and *TET2* mutations with a VAF range of 1–46%. In myeloma patients, *DNMT3A* mutations were present with a VAF range of 2–19%, and *TET2* mutations with a VAF range of 14% to 54% (Appendix A).

### 2.3. HDCT/ASCT

The median interval from diagnosis of the hemato-oncologic malignancy to ASCT was 4.4 months in the CH group and 5 months after diagnosis in the noCH group (Appendix A). Five CH patients (14%) and 22 noCH patients (21%) received a second HDCT/ASCT as part of their relapse treatment. There was no difference regarding neutrophil engraftment after ASCT in both groups (median 11 days) (Appendix A). Interestingly, four CH patients had delayed hematological recovery (11%) in contrast to none in the noCH group (*p* = 0.004). Delayed hematological regeneration, also known as poor graft function, was defined as two or three cytopenias for more than two weeks after day 28 after ASCT [14].

### 2.4. Outcome/Survival Analysis

ASCT therapy was very effective in both groups with complete response rates of 81 to 89% (Table 2). However, more patients with CH (13/36, 36%) than patients without CH (14/106, 17%) relapsed or had progressive disease after ASCT (*p* = 0.017). The median overall survival (OS) did not differ significantly between CH and noCH patients (*p* = 0.16), but there was a trend to reduced OS in patients with CH after 12 months (*p* = 0.06) (Figure 1). Progression-free survival (PFS) was inferior in patients with CH compared to patients without CH (*p* = 0.04). This difference was significant, but with a wide range of intervals until relapse/progression in both groups (from a few weeks to 13.8 years). Secondary malignancies occurred in both groups at low rates, with early occurrence in the CH group. The death rate appeared to be elevated in the CH group (22% vs. 11%, *p* = 0.16).

### 2.5. Survival Analysis in Myeloma, Lymphoma, and AML

In myeloma patients OS and PFS did not differ significantly between the CH and the noCH groups (*p* = 0.26 and *p* = 0.12, respectively) (Figure 2A,B). Complete remission rates were high in myeloma patients with 92% (CH) and 73% (noCH) (Appendix A). One CH patient (8%) and six noCH patients (12%) had relapse or progressive disease, and in both groups there was one patient with a secondary malignancy. All five patients, who died during follow-up, were in the noCH group. The median follow-up was 13.1 months in the CH group and 15.2 months in the noCH group.

In lymphoma patients, OS tended to be shorter in patients with CH (*p* = 0.078) (Figure 2C,D). Moreover, relapse rates were significantly higher in lymphoma patients with CH (*p* = 0.007) (Table 3). Six patients in the CH group (67%) and 32 patients in the noCH group (84%) reached complete remission (*p* = 0.34). Two CH patients (22%) and one noCH patient (3%) developed progressive disease after ASCT (*p* = 0.09). Relapse or progression after ASCT was documented in six (67%) of the CH patients and six (16%) of the noCH patients (*p* = 0.005). The median follow-up was 22.6 months in the CH group and 15.9 months in the noCH group (*p* = 0.45).

In AML patients, the OS and PFS did not differ significantly in the two groups (Figure 2E,F). All AML patients were in complete remission 100 days post ASCT, independent of CH mutation (Appendix A). Relapse rates were elevated in AML patients with CH (43% versus 29%, *p* = 0.48). Secondary malignancies were found in two noCH patients (12%) and in none of the CH patients (*p* = 0.49). Four CH patients (29%) and two noCH patients (12%) died (*p* = 0.37). The median follow-up was 17.8 months in the CH group and 24.6 months in the noCH group (*p* = 0.15).

## 3. Discussion

In recent years, various studies focused on the frequency and impact of clonal hematopoiesis in healthy individuals as well as in patients with hematologic malignancies. As autologous hematopoietic stem cell transplantation following high-dose chemotherapy plays a relevant role for patients with different hematologic malignancies [1,15,16,17], it is of particular interest to evaluate the clinical impact of CH specifically in this context. Here we investigated the phenomenon of CH in patients with different hematologic malignancies (myeloma, lymphoma, AML) who received HDCT/ASCT in standardized treatment protocols in a tertiary university center.

First, we identified CH-associated gene mutations in the peripheral blood of patients who underwent HDCT/ASCT in our center. Gene mutations in *ASXL1*, *DNMT3A*, *TET2*, and *TP53* associated with CH were prevalent in peripheral blood monocytes of many AML patients (45%) and less common in lymphoma (19%) and myeloma (20%) patients. The variant allele frequencies of 1% to 54% indicate an individual range of mutated stem cell clones in different patients with small expansion in some, and massive expansion of one mutated stem cell clone in others. Most patients were in complete remission after ASCT. The frequencies of CH in our cohorts were similar to other studies which found a prevalence of CH of 41.6% in AML in complete remission prior to ASCT [18], 21.6% in myeloma [19], and 29.9% in lymphoma patients [7]. Mutations in *ASXL1*, *DNMT3A*, *TET2, JAK2*, and *TP53* genes are commonly present in AML patients with CH [20]. Thus, our study confirms that CH is more frequent in AML as compared to myeloma and lymphoma patients. This allows the hypothesis that the phenomenon of CH may more frequently promote the development of AML, a myeloid malignancy, as compared to myeloma and lymphoma, both lymphoid malignancies. The investigation of CH by NGS studies was conducted before HDCT in our study. CH may have originated before HDCT/ASCT at various rates in relation to previous cytotoxic therapies, which may have impacted disease progression.

In the outcome and survival analyses of AML patients following HDCT/ASCT, we found no significant differences between the CH and non-CH patients. Similar results were also reported in other studies [5,21]. Considering the high proportion of more than 40% of AML patients with CH in our study and the above-cited literature, it is possible that the clinical effects of CH may be overruled by stronger prognostic parameters in patients with AML (e.g., patient’s age, or the cytogenetic and molecular leukemia-associated risk profile/ELN risk).

Interestingly, lymphoma patients with CH were more likely to have disease relapse and progression following HDCT/ASCT, with significantly reduced PFS and OS in patients with CH. Similar results were reported previously by Gibson et al. in more than 400 lymphoma patients (of those, 29.9% with CH) with an inferior 10-year survival of CH patients after HDCT/ASCT (30.4% vs. 60.9%, *p* < 0.001) [7]. Lackraj et al. found an inferior 5-year OS after HDCT/ASCT (373 patients, 51.8% vs. 59.3%, *p* = 0.018), but no significant difference between CH and non-CH in the risk to relapse or in PFS [22].

For the myeloma patients in our cohort, clinical outcomes were similar in the presence or absence of CH. In contrast, other studies reported an inferior outcome in myeloma patients with CH. In their study in HDCT/ASCT recipients, Mouhieddine et al. found a median OS of 5.3 vs. 7.5 years (*p* = 0.02) and a median PFS of 2.2 vs. 2.6 years (*p* < 0.001) in patients with CH and without CH, respectively [19]. The follow-up in our myeloma cohort was a median 13–15 months. A longer follow-up may have detected differences in outcome in the myeloma cohort.

In conclusion, our study confirms previous results that lymphoma patients with CH have a higher relapse/progression risk in response to HDCT/ASCT. In contrast, the presence of CH may not impact the outcomes in AML patients. For patients with multiple myeloma, larger studies and longer follow-ups are required to further clarify the impact of CH on the outcomes of HDCT/ASCT.

## 4. Materials and Methods

### 4.1. Study Design and Patients

This retrospective single-center study included patients with AML, myeloma, or lymphoma undergoing HDCT/ASCT between 2002 and 2021 at the University Hospital of Bern, with available data for the five common CH mutations *ASXL1*, *DNMT3A*, *JAK2*, *TET2*, and *TP53* at the time of HDCT/ASCT. We documented demographic data, disease characteristics, clinical courses, and molecular results. The primary study endpoint was progression-free survival (PFS) after HDCT/ASCT.

### 4.2. Treatment

Patients received standard chemo-immuno-therapy for their specific hemato-oncologic disease followed by consolidation treatment. In AML patients, treosulfan/melphalan was the standard HDCT [15]. In lymphoma patients, BeEAM (bendamustine, etoposide, cytarabine, and melphalan) was predominantly applied [16]. In myeloma, again, treosulfan and melphalan were predominantly given [17]. Consolidation therapy was followed by HDCT/ASCT in all patients [15,16,17].

### 4.3. Survival

Progression-free survival (PFS) was defined as the time from ASCT to disease progression, relapse, lost to follow-up, or death from any cause. Overall survival (OS) was defined as the time from ASCT to death from any cause.

### 4.4. Molecular Analyses

Genomic DNA was extracted from peripheral blood mononuclear cells (PBMCs) collected on the day of remission assessment before HDCT/ASCT. Next-generation sequencing (NGS) assay was performed on TruSight Oncology 500 (TSO500, Illumina, San Diego, CA, USA). Molecular analysis included a predefined panel of the five genes, *ASXL1*, *DNMT3A*, *JAK2*, *TET2*, and *TP53*. The most frequently mutated genes in clonal hematopoiesis (CH) are *DNMT3A*, *TET2*, and *ASXL1* (DTA) [12,23].

### 4.5. Assessment and Definition of Clonal Hematopoiesis

CH was defined as the presence of mutations in *ASXL1*, *DNMT3A*, *JAK2*, *TET2*, and *TP53* in PBMC isolated from patients before HDCT. The minimal variant allele frequency (VAF) was 1%. Especially in the AML patients, CH had been discriminated from disease-related mutations by taking the clinical and molecular courses into account. Gene mutations which had cleared after HDCT/ASCT were considered disease-related, whereas persisting gene mutations with VAF ≥1% were considered to be CH-associated.

### 4.6. Statistical Analyses

Survival curves (Kaplan–Meyer) and univariate statistical analyses were performed on GraphPad Prism version 8 (GraphPad Software, San Diego, CA, USA). The categorical variables were summarized as frequencies and percentages, and the continuous variables as medians and ranges. The *p*-values were calculated using Fisher’s exact test, the Chi-Square test, or the Mann–Whitney test. Survival analyses were performed using a log-rank (Mantle–Cox) test.

## 5. Conclusions

We identified CH-related gene mutations in the peripheral blood of patients with hematological malignancies undergoing HDCT/ASCT. Mutations in *ASXL1*, *DNMT3A*, *TET2*, and *TP53* genes associated with CH were prevalent in AML (45%) and less frequent in lymphoma (19%) and myeloma (20%). The relapse/progression risk was significantly elevated in CH-positive lymphoma patients, confirming results of previous studies. Post-transplant follow-up strategies in lymphoma patients should be adapted accordingly. In contrast, in AML patients, CH did not have a clinical impact in our and previous studies, with other prognostic factors overruling a possible impact of CH. The situation in multiple myeloma patients has to be further clarified by larger studies with a longer follow-up.

## Figures and Tables

**Figure 1 ijms-26-08021-f001:**
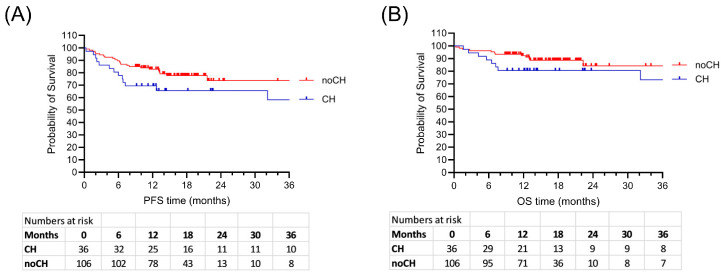
Reduced survival of patients with CH in the entire cohort comprising 64 myeloma, 47 lymphoma, and 31 AML. (**A**) Progression-free survival, *p* = 0.043. (**B**) Overall survival, *p* = 0.163.

**Figure 2 ijms-26-08021-f002:**
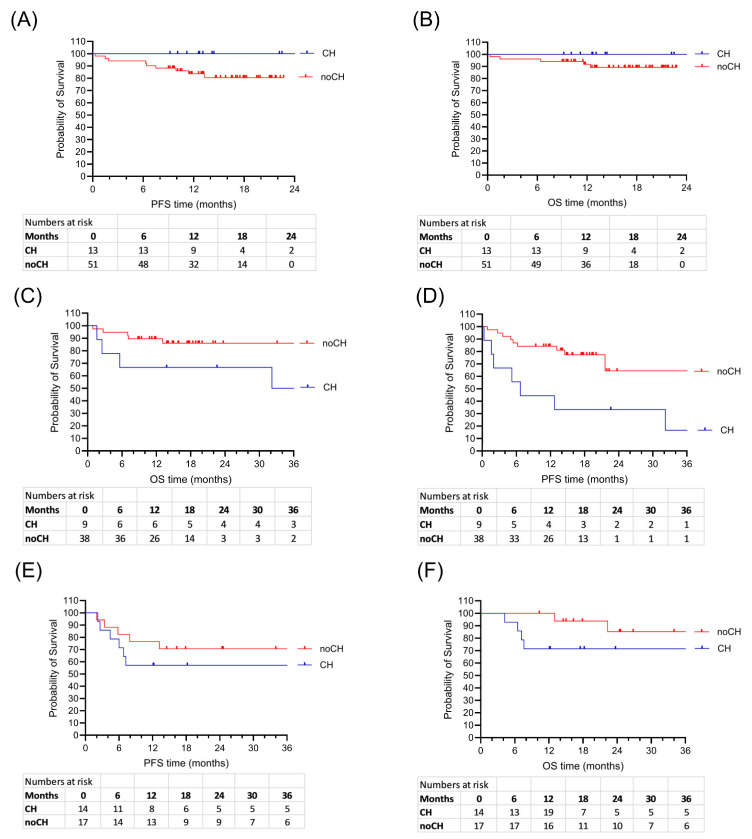
Survival rates after HDCT and ASCT in patients with and without CH gene mutations. Survival rates of 64 myeloma patients: PFS, *p* = 0.12 (**A**) and OS, *p* = 0.26 (**B**); 47 lymphoma patients: PFS, *p* = 0.007 (**C**) and OS, *p* = 0.078 (**D**); 31 AML patients: PFS, *p* = 0.39 (**E**) and OS, *p* = 0.17 (**F**).

**Table 1 ijms-26-08021-t001:** Patient and disease characteristics at diagnosis.

Parameter			CH (n = 36)	NoCH (n = 106)	*p*-Value
Age at diagnosis, median, years (range)			63.5 (33–74)	60.5 (23–75)	0.49
Gender, male, n (%)			21 (58)	72 (68)	0.32
**Myeloma, n (%)**			13 (36)	51 (48)	0.25
	IgG, n (%)		10 (77)	35 (69)	0.74
	IgA, n (%)		2 (15)	9 (18)	>0.99
	IgD, n (%)		0 (0)	1 (2)	>0.99
	Lambda light chain, n (%)		4 (31)	18 (35)	>0.99
	Kappa light chain, n (%)		9 (69)	33 (65)	>0.99
	Light chain only, n (%)		1 (8)	6 (12)	>0.99
	Bone marrow infiltration (%), median (range)		27 (10–80)	50 (3–100)	0.52
	Hypercalcemia, n (%)		2 (15)	13 (25)	0.72
	Kidney failure, n (%)		3 (23)	11 (22)	>0.99
	Creatinine (μmol/L), median (range)		94 (58–195)	82 (53–322)	0.69
	Anemia, n (%)		5 (38)	19 (37)	>0.99
	Hemoglobin (g/L), median (range)		118.5 (83–146)	122 (70–158)	0.99
	Osteolytic lesions, n (%)		11 (85)	37 (73)	0.49
	β2-microglobulin > 3.5 mg/L, n (%)		3 (23)	19 (37)	0.51
	Albumin < 35 g/L, n (%)		7 (54)	10 (20)	**0.03**
	LDH, U/L, median (range)		334 (202–456)	202 (151–436)	0.064
	FISH/cyto- genetics	High risk, n (%)	3 (23)	14 (27)	>0.99
		No high risk, n (%)	6 (46)	22 (43)	>0.99
	(R-)ISS, n (%)	Stage 1	3 (23)	13 (25)	>0.99
		Stage 2	3 (23)	23 (45)	0.21
		Stage 3	5 (38)	10 (20)	0.16
	Revised Mayo Stage 3, n (%)		1 (8)	2 (4)	0.50
**Lymphoma, n (%)**			9 (25)	38 (36)	0.31
	Hodgkin lymphoma, n (%)		1 (11)	5 (13)	>0.99
	DLBCL, n (%)		4 (44)	17 (45)	>0.99
	Transformed, n (%)		2 (50)	6 (35)	0.62
	Mantle cell lymphoma, n (%)		1 (11)	11 (29)	0.41
	Follicular lymphoma, n (%)		0 (0)	1 (3)	>0.99
	Peripheral T-cell lymphoma, n (%)		0 (0)	3 (8)	>0.99
	Angioimmunoblastic T-cell Lymphoma (AITL), n (%)		3 (33)	0 (0)	**0.005**
	Burkitt lymphoma, n (%)		0 (0)	1 (3)	>0.99
	Stage (Ann Arbor), n (%)	I	0 (0)	0 (0)	>0.99
		II	1 (11)	4 (11)	>0.99
		III	1 (11)	6 (16)	>0.99
		IV	6 (67)	24 (63)	>0.99
	Bone marrow infiltration, n (%)		2 (22)	13 (34)	0.69
**AML, n (%)**			14 (39)	17 (16)	**0.009**
	FAB, n (%)	M0	1 (7)	1 (6)	>0.99
		M1	2 (14)	7 (41)	0.13
		M2	7 (50)	5 (29)	0.28
		M3	0 (0)	0 (0)	>0.99
		M4	3 (21)	2 (12)	0.63
		M4Eo	1 (7)	2 (12)	>0.99
	De novo AML Secondary AML		12 (86) 2 (14)	16 (94) 1 (6)	0.58
	Leukocytes (G/L), median (range)		11.2 (1.35–272)	12 (0.64–214)	0.99
	Thrombocytes (G/L), median (range)		98 (38–128)	35 (5–169)	0.075
	Hemoglobin (g/L), median (range)		94.5 (60–130)	84 (63–129)	0.28
	Peripheral blasts (%), median (range)		60 (5–95)	55 (13.5–98)	0.95
	Bone marrow infiltration (%), median (range)		85 (20–95)	90 (20–100)	0.36
	LDH (U/L), median (range)		740 (270–2514)	860 (162–2225)	0.63
	Mutations, n (%)	Mut. *NPM1* and wt *FLT3*	4 (29)	6 (35)	>0.99
		Mut. *NPM1* and mut. *FLT3*	3 (21)	4 (24)	>0.99
		Mut. *FLT3* and wt *NPM1*	2 (14)	0 (0)	>0.99
		Mut. *IDH2*	2 (14)	4 (24)	0.19
		t(8;21)/*RUNX1-RUNX1T1*	1 (7)	1 (6)	0.66
		inv(16)/*CBFB-MYH11*	1 (7)	1 (6)	>0.99
		Mut. *SRSF2*	2 (14)	3 (18)	>0.99
		Mut. *NRAS*	2 (14)	1 (6)	>0.99

FISH in 4 CH and 15 noCH pts unknown, (R)-ISS in 1 CH and 3 noCH pts unknown, Ann Arbor in 1 CH and 4 noCH pts unknown; CH: clonal hematopoiesis of indeterminate potential; noCH: patients without CH; *p*-value for difference between CH and noCH group; LDH: lactate dehydrogenase; FISH/cytogenetics high-risk mutations: t(4;14), t(14;16), del(17p), t(14;20), +1q; (R-)ISS: (revised) international staging system; DLBCL: diffuse large B-cell lymphoma; AML: acute myeloid leukemia; FAB: French–American–British; Mut.: mutated; wt: wild type.

**Table 2 ijms-26-08021-t002:** Clinical outcomes in the entire cohort (lymphoma, myeloma, and AML).

Parameter		CH (n = 36)	NoCH (n = 106)	*p*-Value
Remission after ASCT, n (%)				
	CR (complete remission)	32 (89)	86 (81)	0.44
	VGPR (very good partial remission)	0 (0)	4 (4)	0.57
	PR (partial remission)	1 (3)	9 (8)	0.45
	SD (stable disease)	0 (0)	1 (1)	>0.99
	PD (progressive disease)	2 (6)	2 (2)	0.26
Relapse/progression after ASCT, n (%)		13 (36)	17 (16)	**0.017**
	Months until relapse/progression after ASCT, median (range)	6 (0.3–165)	6 (2–22)	0.54
Secondary malignancy, n (%)		2 (6)	4 (4)	0.64
	Months until second. malignancy after ASCT, median (range)	25 (24–26)	60 (12–226)	>0.99
Death, n (%)		8 (22)	12 (11)	0.16
	Months until death after ASCT, median (range)	6 (1.6–32)	7 (0.3–22)	0.85
	Death related to the underlying disease or therapy, n (%)	7 (88)	11 (92)	>0.99
Follow-up, months, median (range)		14.3 (1.6–220.7)	16.5 (0.3–248.4)	0.84

CH: clonal hematopoiesis of indeterminate potential; noCH: patients without CH; ASCT: autologous stem cell transplantation.

**Table 3 ijms-26-08021-t003:** Clinical outcomes in lymphoma patients.

	Parameters	CH (n = 9)	NoCH (n = 38)	*p*-Value
Remission status after ASCT, n (%)				
	CR (complete remission)	6 (67)	32 (84)	0.34
	PR (partial remission)	0 (0)	3 (8)	>0.99
	SD (stable disease)	0 (0)	0 (0)	>0.99
	PD (progressive disease)	2 (22)	1 (3)	0.09
Relapse/progression after ASCT, n (%)		6 (67)	6 (16)	**0.005**
	Months until relapse/progression after ASCT, median (range)	5.9 (0.3–165)	5.8 (3.6–22)	0.82
Secondary malignancy, n (%)		1 (11)	1 (3)	0.35
	Months until second malignancy after ASCT, median (range)	24.4 (-)	15.9 (-)	-
Death, n (%)		4 (44)	5 (13)	0.054
	Months until death after ASCT, median (range)	4 (1.6–32)	7 (0.9–13)	0.90
	Death related to the underlying disease or therapy, n (%)	3 (75)	4 (80)	>0.99
Follow-up, months, median (range)		22.6 (1.6–180.0)	15.9 (0.9–109.5)	0.45

CH: Clonal hematopoiesis of indeterminate potential; noCH: patients without CH; *p*-value for difference between CH and noCH groups; ASCT: autologous stem cell transplantation.

## Data Availability

Data are contained within the article. No data are deposited elsewhere.

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
