# Peer review of "Clonal Hematopoiesis and Outcomes After High-Dose Chemotherapy and Autologous Stem Cell Transplantation in Patients with AML, Myeloma, and Lymphoma"

_ijms, 2025, doi:10.3390/ijms26168021_

Round 1

Reviewer 1 Report

Comments and Suggestions for Authors
  1. Please clarify CH Definition. CH definitions vary between studies, please make sure the variant allele frequency (VAF) threshold and cutoff, the specific genes included in the panel, and whether germline variants were filtered out.
  2. Please clarify CH Time Point, CH status before transplantation and after potential treatment-related clonal selection.
  3. Questions to Survival Analysis: the manuscript combines survival data from patients with AML, myeloma, and lymphoma without stratification or statistical adjustment. These diseases have vastly different biology, treatment protocols, and prognoses. Pooling them may introduce significant confounding, especially when evaluating the impact of CH on survival.
  4. The manuscript does not clearly indicate the comparator groups used in statistical analyses (e.g., CH+ vs CH−). The sample size is too small to help assess clinical relevance beyond p-values.
  5. The results and discussion merge findings from AML, myeloma, and lymphoma, without adequately highlighting disease-specific outcomes. CH may have different clinical implications in different disease contexts.
  6. Improve Language Clarity

Some sentences are overly long, contain redundant clauses, or use ambiguous terms. Please revise complex sentences for conciseness and clarity. A language polish by a native speaker or editorial service is recommended.

  1. Supplementary Tables S1–S6 are informative but are not consistently referenced in the main text.
Comments on the Quality of English Language

Improve Language Clarity

Some sentences are overly long, contain redundant clauses, or use ambiguous terms. Please revise complex sentences for conciseness and clarity. A language polish by a native speaker or editorial service is recommended.

Author Response

Comments and Suggestions for Authors

  1. Please clarify CH Definition. CH definitions vary between studies, please make sure the variant allele frequency (VAF) threshold and cutoff, the specific genes included in the panel, and whether germline variants were filtered out.

Response: CH definition in chapter 2.5. Assessment and Definition of Clonal Hematopoiesis. CH was defined as the presence of mutations in ASXL1, DNMT3A, JAK2, TET2 and TP53 at a variant allele frequency (VAF) of at least 1% in PBMC of patients before HDCT/ASCT. In AML patients, CH was distinguished from leukemia-associated mutations. Gene mutations that were cleared after HDCT/ASCT were considered disease-related, whereas persisting gene mutations with VAF levels of ≥1%, were considered CH-related. We stated this more clearly in the revised methods section. CH related gene mutations were screened in PBMC collected at remission assessment before HDCT/ASCT. Germline variants were not screened.

  1. Please clarify CH Time Point, CH status before transplantation and after potential treatment-related clonal selection.

Response: We analyzed CH status on the day of remission assessment before HDCT/ASCT. We stated this more clearly in the revised methods section. Molecular analysis in chapter 2.4. Genomic DNA was extracted from peripheral blood mononuclear cells collected on the day of remission assessment before HDCT/ASCT. Next generation sequencing (NGS) was performed on TruSight Oncology 500 (TSO500, Illumina, San Diego, CA, USA). Molecular analysis included a predefined panel of the five genes ASXL1, DNMT3A, JAK2, TET2 and TP53.

  1. Questions to Survival Analysis: the manuscript combines survival data from patients with AML, myeloma, and lymphoma without stratification or statistical adjustment. These diseases have vastly different biology, treatment protocols, and prognoses. Pooling them may introduce significant confounding, especially when evaluating the impact of CH on survival.

Response: Survival was analyzed in patients with CH with versus without CH (Figure 1, Table 2), followed by a stratified analysis in the different disease subgroups (Figure 2, Table 3, Table S5, S6).

  1. The manuscript does not clearly indicate the comparator groups used in statistical analyses (e.g., CH+ vs CH−). The sample size is too small to help assess clinical relevance beyond p-values.

Response: The comparator groups are CH versus no CH in all tables. The small sample sizes precluded subgroup analysis.

  1. The results and discussion merge findings from AML, myeloma, and lymphoma, without adequately highlighting disease-specific outcomes. CH may have different clinical implications in different disease contexts.

Response: We agree with this important conclusion. In lymphoma patients, the relapse/ progression risk post-ASCT was significantly increased in patients with CH, confirming results of previous studies. In contrast, in AML patients, CH did not have a clinical impact in our and previous studies, with other prognostic factors overruling a possible impact of CH. The situation in multiple myeloma patients has to be further clarified by larger studies with a longer follow-up.  

  1. Supplementary Tables S1–S6 are informative but are not consistently referenced in the main text.

Response: References to Tables S1-S6 have been realigned in the revised version of the manuscript as requested by the reviewer.

Comments on the Quality of English Language

  1. Improve Language Clarity

Some sentences are overly long, contain redundant clauses, or use ambiguous terms. Please revise complex sentences for conciseness and clarity. A language polish by a native speaker or editorial service is recommended.

Response: Language was polished by our internal editorial service.

We thank the reviewer for the detailed critical comments. We have addressed all the issues. We believe that the manuscript has been substantially improved.

Submission Date

01 July 2025

Date of this review

14 Jul 2025 11:33:52

Date of revision 7 August 2025

Reviewer 2 Report

Comments and Suggestions for Authors

The authors showed that mutations in ASXL1, DNMT3A, TET2, and TP53, associaten with CH, were common in acute myeloid leukemia (AML) (45%) and less frequent in lymphomas (19%) and multiple myeloma (20%). In lymphoma patients, the presence of CH significantly increased the risk of relapse or disease progression after ASCT, whereas in AML patients, CH had no clinical significance, and in multiple myeloma patients, it requires further clarification in larger studies with longer follow-up. This is an interesting study, but I have a few comments:

  1. In order to clearly present and distinguish primary mutations (befor treatment) from those associated with clonal hematopoiesis (CH), please supplement the patient data with an analysis of the mutations in the studied genes at the time of diagnosis of AML, lymphoma, or myeloma. This is justified for all genes, but especially for IDH1/IDH2, ASXL1, and TP53, which are often observed as accompanying alterations in these malignancies.
  2. The authors should clearly state whether all patients were in complete remission of the disease on day 100 after auto-HSCT. If this was not the case — especially in AML — it will be difficult to distinguish primary mutations from those associated with CH.
  3. In all patients, CH mutations did not occur at a low allelic fraction — there were patients with a fraction ranging from 30% to over 40%. The authors are asked to clarify this situation, particularly in relation to AML, and to provide a comment on this in the discussion.
  4. CH typically presents with a low variant allele frequency (VAF), around 1–20%. In overt malignancy, the clone is often dominant (VAF >20–30% or higher). In patients with multiple myeloma (MM), CH is detected in approximately 10–25% of cases at diagnosis, and this proportion increases to 20–30% after treatment, particularly following chemotherapy or ASCT. Could the authors comment on their data in MM — both at diagnosis and post-ASCT?
  5. In the discussion, the authors should comment on the fact that CH in MM can be both pre-existing (age-related) and therapy-induced.
  6. AML very often develops from a pre-existing CH clone that acquires additional transforming mutations (e.g., NPM1, FLT3, IDH1/2, RUNX1). Could the authors comment in discussion on how to distinguish CH from leukemia? After all, in some patients even after AML is cured, CH persists — because DNMT3A/TET2/ASXL1 mutations remain in hematopoietic stem cells that survived therapy.
  7. In the case of lymphomas, the situation is simpler — CH affects hematopoietic (myeloid) precursors, while lymphoma arises from lymphocytes. However, the data on mutation analysis before treatment should also be presented.

Author Response

Comments and Suggestions for Authors

The authors showed that mutations in ASXL1, DNMT3A, TET2, and TP53, associaten with CH, were common in acute myeloid leukemia (AML) (45%) and less frequent in lymphomas (19%) and multiple myeloma (20%). In lymphoma patients, the presence of CH significantly increased the risk of relapse or disease progression after ASCT, whereas in AML patients, CH had no clinical significance, and in multiple myeloma patients, it requires further clarification in larger studies with longer follow-up. This is an interesting study, but I have a few comments:

Response: We thank the reviewer for the detailed critical comments. We have addressed all the issues. We believe that the manuscript has been substantially improved.

  1. In order to clearly present and distinguish primary mutations (befor treatment) from those associated with clonal hematopoiesis (CH), please supplement the patient data with an analysis of the mutations in the studied genes at the time of diagnosis of AML, lymphoma, or myeloma. This is justified for all genes, but especially for IDH1/IDH2, ASXL1, and TP53, which are often observed as accompanying alterations in these malignancies.

Response: Disease characteristics at diagnosis are presented in Table 1. At diagnosis, six AML patients had IDH2 mutations, five AML patients had SRSF2 mutations, and three AML patients had NRAS mutations. CH associated mutations are presented in Table S1 and S2. In peripheral blood cells of 14 AML patients there were two ASXL1 mutations, nine DNMT3A mutations, five TET2 mutations. We added this information in the revised results section as suggested.

  1. The authors should clearly state whether all patients were in complete remission of the disease on day 100 after auto-HSCT. If this was not the case — especially in AML — it will be difficult to distinguish primary mutations from those associated with CH.

Response: In all AML patients the response evaluation suggested morphologic and flow-cytometric absence of disease (AML) 100 days post ASCT. In Lymphoma, 6 CH patients (67%) and 32 noCH patients (84%) were in CR (Table 3). In myeloma, 12 CH patients (92%) and 37 noCH patients (73%) were in CR, as depicted in Table S5. In summary, whereas all AML patients are in ongoing CR at day 100 after ASCT, some lymphoma and myeloma patients have not achieved or maintained CR at the day 100 response evaluation. We stated this in the revised results section.

  1. In all patients, CH mutations did not occur at a low allelic fraction — there were patients with a fraction ranging from 30% to over 40%. The authors are asked to clarify this situation, particularly in relation to AML, and to provide a comment on this in the discussion.

Response: The variant allele frequencies of 1% to 54% indicate an individual range of mutated stem cell clones in different patients with small expansion in some, and massive expansion of one mutated stem cell clone in others. This comment was added to the discussion.

  1. CH typically presents with a low variant allele frequency (VAF), around 1–20%. In overt malignancy, the clone is often dominant (VAF >20–30% or higher). In patients with multiple myeloma (MM), CH is detected in approximately 10–25% of cases at diagnosis, and this proportion increases to 20–30% after treatment, particularly following chemotherapy or ASCT. Could the authors comment on their data in MM — both at diagnosis and post-ASCT?

Response: CH assessment was performed at the status assessment before HDCT/ASCT in this study. In myeloma patients, DNMT3A mutations were present with VAF range of 2-19%, TET2 mutations with VAF range of 14-54%. We stated this in the revised results section.

  1. In the discussion, the authors should comment on the fact that CH in MM can be both pre-existing (age-related) and therapy-induced.

Response: This is an important point made by the reviewer. Because CH status was not analyzed at diagnosis, we cannot differentiate whether CH is caused by therapy or was already pre-existing/age-related. We added a comment on this limitation in the discussion.

  1. AML very often develops from a pre-existing CH clone that acquires additional transforming mutations (e.g., NPM1, FLT3, IDH1/2, RUNX1). Could the authors comment in discussion on how to distinguish CH from leukemia? After all, in some patients even after AML is cured, CH persists — because DNMT3A/TET2/ASXL1 mutations remain in hematopoietic stem cells that survived therapy.

Response: We thank the reviewer for bringing up this important issue, and we agree. AML-related gene mutations that were cleared after HDCT/ASCT were considered disease related. DNMT3A mutations that were cleared after intensive chemotherapy and ASCT were considered leukemia-related, whereas persisting DNMT3A mutations with VAF levels of ≥1%, despite achieving hematologic CR and despite the clearance of other coincidental mutations at a follow-up, were considered to be CH. We stated this more clearly in the revised methods section.

  1. In the case of lymphomas, the situation is simpler — CH affects hematopoietic (myeloid) precursors, while lymphoma arises from lymphocytes. However, the data on mutation analysis before treatment should also be presented.

Response: We agree. Results of NGS analysis at diagnosis were not available in the majority of patients. We have presented the CH status before ASCT in Table S1.

Submission Date

01 July 2025

Date of this review

21 Jul 2025 18:20:32

Date of revision 07 August 2025
